# Impacts of Proanthocyanidin Binding on Conformational and Functional Properties of Decolorized Highland Barley Protein

**DOI:** 10.3390/foods12030481

**Published:** 2023-01-19

**Authors:** Juan Li, Xin Zhang, Wenju Zhou, Zhaoxin Tu, Xijuan Yang, Jing Hao, Feng Liang, Zhengxing Chen, Yan Du

**Affiliations:** 1National Engineering Research Center of Cereal Fermentation and Food Biomanufacturing, Jiangnan University, Wuxi 214122, China; 2Jiangsu Provincial Engineering Research Center for Bioactive Product Processing, Jiangnan University, Wuxi 214122, China; 3Qinghai Tianyoude Technology Investment Management Group Co., Ltd., Qinghai Engineering Technology Research Institute for Comprehensive Utilization of Highland Barley Resources, Xining 810016, China; 4Qinghai Tibetan Plateau Key Laboratory of Agric-Product Processing, Qinghai Academy of Agricultural and Forestry Sciences, Xining 810016, China

**Keywords:** functional properties, decolorized protein, highland barley, proanthocyanidin, structural changes

## Abstract

The impacts of interaction between proanthocyanidin (PC) and decolorized highland barley protein (DHBP) at pH 7 and 9 on the functional and conformational changes in DHBP were investigated. It was shown that PC strongly quenched the intrinsic fluorescence of DHBP primarily through static quenching. PC and DHBP were mainly bound by hydrophobic interactions. Additionally, free sulfhydryl groups and surface hydrophobicity obviously decreased in DHBP after combining with PC. The zeta potential of DHBP–PC complexes at pH 7 increased significantly. A change in the structure of DHBP was caused by interactions with PC, resulting in an increase in the number of β-sheets, a decrease in the number of α-helixes, and a spectral shift in the amide Ⅱ band. Furthermore, the presence of PC enhanced the foaming properties and antioxidant activity of DHBP. Overall, this study suggests that DHBP–PC complexes at pH 7 could be designed as a stable additive, and illustrates the potential applications of DHBP–PC complexes in the food industry.

## 1. Introduction

Highland barley is called hull-less barley or naked barley because its inner and outer glumes are separated from caryopsis when harvested [1]. In Tibet, it is a major crop and a major staple food. It is rich in proteins, fibers, vitamins, β-glucans, arabinoxylans, polyphenols, and flavonoids [2,3,4]. Furthermore, it contains lysine (0.360 g/100 g) and threonine (0.360 g/100 g), which are considered limiting amino acids [5]. Highland barley protein is well documented to reduce the prevalence of cardiovascular diseases and atherosclerosis [5]. However, the data regarding the structural and functional properties of highland barley protein are lacking; most researchers have focused on their derivatives, that is, the peptides [5]. Moreover, natural highland barley proteins cannot meet the different needs of food processing. For example, a large amount of wheat flour and gluten powder are often added in the processing of highland barley noodles due to the fact that highland barley protein cannot form gluten network structures [5]. Studies revealed that its structural and functional properties, as well as its biological activities, can be improved by modifying the properties of highland barley proteins to broaden their market application.

Proanthocyanidin (PC) is one of the major polyphenols in highland barley. PC has a wide range of health benefits, including antioxidant, antibacterial, antiviral, anticarcinogenic, anti-inflammatory, antiallergic, and vasodilatory actions [6]. Moreover, it is used as an additive in food formulations to enhance the microbial stability, foamability, oxidative stability, heat stability, and shelf life of several food products [6]. Nevertheless, PC application is restricted because of its susceptibility to breakdown by light and oxygen [7]. Furthermore, PC has extremely low bioavailability due to its chemical instability in gastrointestinal fluids [8]. Studies revealed that combining polyphenols with proteins could improve bioavailability [9]. Consequently, it might be possible to improve the functional properties of PC by binding it to proteins.

The interactions of polyphenols and proteins frequently appear in food processing [10]. Previous studies have indicated that the combination of proteins and polyphenols could alter the structural, nutritional, functional, and digestive properties of proteins [9]. For example, soy proteins accompanying anthocyanins could change the secondary structure and improve the foaming properties and emulsifying properties of soy proteins [10]. The surface hydrophobicity was significantly reduced, and the emulsification, foaming, and digestibility (in vitro) of the pea protein isolate were enhanced after noncovalent binding between epigallocatechin-3-gallate, resveratrol, and chlorogenic acid [11]. The binding of rice proteins with anthocyanins could alter the structure, modify morphological characteristics, enhance the functional properties, and improve the antioxidant ability of rice proteins [12]. In our previous studies, the structural and functional characteristics (foaming properties, emulsifying properties, and antioxidant ability) of highland barley proteins were improved by interacting with salidroside [13]. However, comprehensive theoretical studies on the interactions of PC with highland barley proteins have not been reported.

Hence, this study explored the interaction between PC and highland barley protein at different pH values. First, the color of highland barley protein extracted using alkali and acid precipitation was brown, which might be due to the combination of various phenolic substances under alkaline conditions. The decolorized highland barley protein (DHBP) was taken as the research object to reduce interference. Second, the fluorescence quenching at different temperatures and the binding parameters were calculated. Third, Fourier transform infrared spectroscopy (FTIR), circular dichroism (CD), turbidity, surface hydrophobicity (*H*_0_), zeta potential, and particle size were measured. Finally, the antioxidant activities of the DHBP–PC complexes were evaluated. This study can provide a theoretical basis for the application and development of highland barley protein.

## 2. Materials and Methods

### 2.1. Materials

Kunlun 14 highland barley flour was provided by Qinghai Tianyoude Technology Investment Management Group Co., Ltd. (Xining, China). PC (purity > 95%) was purchased from Shanghai Maclin Biochemical Technology Co., Ltd. (Shanghai, China). Ortho-phthaldialdehyde (OPA), 1,1-diphenyl-2-pycrylhydrazyl (DPPH), 2,2-azino-bis (3-ethylbenzothiazoline-6-sulfonic acid) (ABTS), and 2,4,6-tris (2-pyridyl)-s-triazine (TPTZ) were obtained from Sigma (St. Louis, MO, USA). Other reagents were purchased from Sinopharm Chemical Co., Ltd. (Shanghai, China).

### 2.2. Preparation of DHBP

Highland barley flour was dispersed in deionized water (1:25, *m*/*v*). The dispersion was adjusted to pH 11.0 and then stirred at 40 °C for 2 h. The suspension was centrifuged at 4800× *g* for 10 min. Then, hydrogen peroxide (3%, *v*/*v*) was added to the suspension to decolorize it, and it was kept at 40 °C for 60 min [14]. After that, the supernatant was adjusted to the isoelectric point (pH 4.5) and centrifuged (4800× *g*, 10 min). The sediment was washed several times with deionized water and freeze-dried.

### 2.3. Preparation of DHBP–PC Complexes

The DHBP–PC complexes were prepared using a slightly modified method proposed by Li et al. [12]. DHBP powder was dispersed in deionized water (10 mg/mL), and the DHBP dispersion was adjusted to pH 12.0. The protein solutions were stirred at room temperature for 1.5 h. Then, the protein solutions were slowly neutralized to pH 7 and centrifuged (4800× *g*, 10 min) to obtain a DHBP solution (pH 7). The DHBP–PC complexes were formed at pH 7 with different concentrations of PC (0, 0.08, 0.16, 0.32, and 0.64 mmol/L) at 25 °C for 2 h.

The protein solutions (pH 12) were slowly adjusted to pH 9 and centrifuged (4800× *g*, 10 min) to obtain a DHBP solution (pH 9). The DHBP–PC complexes were formed at pH 9 with different concentrations of PC (0, 0.08, 0.16, 0.32, and 0.64 mmol/L) at 25 °C for 2 h.

### 2.4. Determination of PC Contents

Total PC contents in DHBP–PC complexes were measured using the Folin phenol method [15]. The dialyzed DHBP–PC complexes were appropriately diluted and measured at 760 nm using a spectrophotometer (UV–3200, Mapada, Shanghai, China). The PC contents were calculated according to the prepared calibration curve (*Y* = 3.4013*X* + 0.0364, *R*^2^ = 0.9994).

### 2.5. Turbidity

The turbidity of DHBP–PC complexes was measured at 600 nm using a spectrophotometer (UV–3200, Mapada).

### 2.6. Particle Size and Zeta Potential

The particle size and zeta potential of DHBP–PC complexes were obtained using a Zetasizer Nano ZS instrument (Malvern, UK). The DHBP–PC complex solutions were slowly injected into the sample cell. All measurements were taken at 25 °C using the refractive indices of proteins (1.45) and deionized water (1.33).

### 2.7. Surface Hydrophobicity

The *H*_0_ of DHBP–PC complexes was measured using 1-anilino-8-naphthalene sulfonate (ANS) as a hydrophobic probe [16]. The DHBP–PC complexes solutions were diluted to serial concentrations (0.03–1.0 mg/mL). Then, 25 μL of 8 mmol/L ANS solution was added to 2 mL of the diluted DHBP–PC complex solutions. An excitation wavelength (390 nm) and an emission wavelength (470 nm) were used to measure the fluorescence intensity.

### 2.8. Fluorescence Spectroscopy Analysis

#### 2.8.1. Fluorescence Spectroscopy

The fluorescence spectra were determined using a spectrofluorimeter (F-7000, Hitachi, Tokyo, Japan) at three different temperatures (298, 303, and 308 K). The fluorescence emission spectra (300–450 nm) of DHBP–PC complexes were determined by exciting the protein at 290 nm.

#### 2.8.2. Fluorescence Quenching

The analysis of fluorescence quenching spectra was applied to elucidate the quenching mechanism. The Stern–Volmer equation was used to determine the type of fluorescence quenching [17]:(1)$$\frac{F_{0}}{F}=1+K_{q}\tau_{0}\left[PC\right]=1+K_{SV}$$
where *F*_0_ (PC absence) and *F* (PC presence) denote the fluorescence intensity of DHBP; *K*_sv_ refers to the quenching constant; *K_q_* is the quenching rate constant of the biological macromolecule; and *τ*_0_ is the average lifetime of the molecule without any quencher (*τ*_0_ = 10^−8^ s) [18]. [*PC*] represents the concentration of PC.

In the case of static quenching, the following Formula (4) was used to calculate the binding constants (*K*_a_) and the number of binding sites (*n*) on DHBP [19]:(2)$$\log\left(\frac{F_{0}-F}{F}\right)=\log K_{a}+n\log\left[PC\right]$$

Based on the fluorescence quenching of DHBP by PC at three different temperatures, the thermodynamic parameters, including enthalpy change (Δ*H*), entropy change (Δ*S*), and Gibbs free energy (Δ*G*), were calculated according to the Van’t Hoff equation [20]:(3)$$\ln K_{a}=-\frac{\Delta H}{RT}+\frac{\Delta S}{R}$$
(4)ΔG=ΔH−TΔS
where *R* is the gas constant [8.314 J/(mol·K)]; *T* represents the temperature; and *K_a_* represents the binding constant between proteins and polyphenols at the corresponding temperature.

### 2.9. Structural Characteristics

#### 2.9.1. Determination of the Contents of Free Sulfhydryl and Amino Groups

The free sulfhydryl contents were measured after reacting with 5,5′-dithiobis (2-nitrobenzoic acid) using a spectrophotometer (UV–3200, Mapada) [21].

The free amino group contents of DHBP–PC complexes were determined by the OPA method with some modifications [22]. The DHBP–PC complex solution (1 mg/mL, 200 mL) and OPA reagent (4 mL) were mixed and then reacted at 35 °C for 2 min. Subsequently, the absorbance at 340 nm was determined using a UV–3200 spectrophotometer (Mapada). The free amino group content was calculated using a calibration curve (*Y* = 3.1858*X* + 0.2021, *R*^2^ = 0.997) of L-leucine as a standard.

#### 2.9.2. CD Spectroscopy

The CD spectra were recorded using a CD spectrometer (Chirascan V100, Applied Photophysics Ltd., Leatherhead, UK) by a previously described method [12]. The DHBP–PC complex solutions were diluted 20 times and then scanned in a wavelength range of 190–250 nm.

#### 2.9.3. FTIR

FTIR spectra were determined at room temperature using a Nicolet IS10 FTIR spectrometer (Madison, WI, USA). An appropriate amount of DHBP–PC complexes were mixed with KBr and laminated. The spectra of DHBP–PC complexes were scanned in the range of 4000–400 cm^−1^ with 32 scans.

### 2.10. Functional Properties Measurement

#### 2.10.1. Foaming Property

The foaming capacity (FC) and foaming stability (FS) of DHBP–PC complexes were measured following the method of Du et al. [14].

#### 2.10.2. Emulsifying Property

The emulsifying activity index (EAI) and emulsifying stability index (ESI) of DHBP and DHBP–PC complexes were measured according to the method of Du et al. [13].

#### 2.10.3. Antioxidant Activity

The ABTS radical scavenging activity of DHBP–PC complexes was measured by the method proposed by Du et al. [13].

The DPPH radical scavenging activity of DHBP–PC complexes was measured by the method proposed by Du et al. [13].

The ferric reducing antioxidant power (FRAP) method was adapted from a previous method with some modifications [23]. The FRAP working solution was prepared by mixing 10 mmol/L TPTZ in 40 mmol/L HCl, 20 mmol/L FeCl_3_, and 0.3 mol/L acetate buffer (pH 3.6) at a ratio of 1:1:10 (*v*/*v*/*v*), and was then kept at 37 °C for 1 h. Then, 36 μL of samples and 1.5 mmol/L Fe_2_SO_4_ standards were added to 270 μL of the FRAP reagent. After that, the mixtures were incubated at 37 °C for 8 min and the absorbance was recorded at 593 nm.

### 2.11. Statistical Analysis

All the experiments were performed in triplicate. The results are expressed as mean ± standard deviation. One-way analysis of variance and Duncan’s test at *p* < 0.05 were used for statistical analysis.

## 3. Results and Discussion

### 3.1. Interactions of DHBP with PC

As shown in Table 1, the binding contents of PC on DHBP increased significantly with the increased reaction concentration of PC (*p* < 0.05), indicating the successful grafting of PC on DHBP molecules. Previous reports revealed that the content of polyphenols bound to soybean protein increased gradually with the increased additive levels of polyphenols, which was in agreement with our results [10]. The myofibrillar protein–gallic acid complex exhibited a similar behavior [24]. In addition, the content of PC retained in the DHBP–PC complexes at pH 7 was significantly higher than that of the DHBP–PC complexes at pH 9 (*p* < 0.05), which might be caused by intrinsic attributes of DHBP [25].

### 3.2. Turbidity and Particle Size

#### 3.2.1. Turbidity

As shown in Figure 1a, the turbidity of DHBP–PC complexes formed at pH 7 increased with the increased concentration of PC. The turbidity of DHBP–PC complexes formed at pH 9 did not initially change substantially. The formation of turbidity is a complex process influenced by several factors, including particle diameters and particle interactions [26]. The increased turbidity could be related to the interactions between DHBP and PC, causing the complex to microaggregate [26]. Furthermore, the turbidity of DHBP–PC complexes at pH 7 was significantly higher than that of the DHBP–PC complexes at pH 9 (*p* < 0.05), which might be caused by the solubility difference of DHBP in different pH environments.

#### 3.2.2. Particle Size

The particle size and polydispersity index (PDI) were measured to analyze DHBP–PC complexes and support the turbidity. As shown in Figure 1b, the particle size of DHBP–PC complexes forming at pH 7 changed insignificantly (*p* > 0.05). No significant changes in the PDI of DHBP–PC complexes at pH 7 were observed at low concentrations (0–0.16 mmol/L). However, the PDI for DHBP–PC complexes at pH 7 increased significantly at high concentrations of PC (0.32–0.64 mmol/L). Meanwhile, the particle size of the DHBP–PC complexes formed at pH 9 reduced with increasing concentrations of PC, which could be mainly due to the stronger interactions between proteins and polyphenols [27]. The PDI of DHBP–PC complexes at pH 9 increased significantly after combining with PC, suggesting that DHBP–PC complexes at pH 9 were widely dispersed. The results suggested that the stability of DHBP–PC complexes formed at pH 9 decreased with the increased concentration of PC. In addition, compared with DHBP–PC complexes at pH 7, the higher particle size and PDI of DHBP–PC complexes at pH 9 indicated that DHBP–PC complexes at pH 9 had lower stability.

### 3.3. Surface Properties

#### 3.3.1. Zeta Potential

The zeta potential of complexes is a useful indicator of dispersion stability [28]. As shown in Figure 1c, the zeta potential value of DHBP was –29.90 ± 0.56 mV at pH 7, while the zeta potential of DHBP was –32.50 ± 0.80 mV at pH 9. Furthermore, PC had a highly negative zeta potential (–32.53 ± 2.79 mV). The absolute zeta potential of DHBP–PC complexes formed at pH 7 increased significantly with the increasing concentration of PC. The increase in absolute zeta potential provided the sufficient repulsive force to keep particles away from each other, leading to stable complexes [10]. Furthermore, the result indicated that DHBP molecules might be surrounded by PC and then their surface charge distribution might be altered, further enhancing the interparticle electrostatic repulsion and improving the stability of dispersions [27]. Furthermore, the increase in absolute zeta potential values was ascribed to the fact that PC changed the secondary structure of DHBP [29]. The absolute zeta potential of DHBP–PC complexes formed at pH 9 decreased significantly with increasing concentration of PC. The lower absolute zeta potential further confirmed the increase in the particle size of DHBP–PC complexes and PDI. This phenomenon occurred because some PC combined with DHBP and reduced the total negative charge on the outer colloidal particle surfaces [8]. Moreover, the absolute zeta potential of DHBP-bound PC at pH 7 was higher than that at pH 9. In brief, the DHBP–PC complexes at pH 7 were more stable than DHBP–PC complexes at pH 9.

#### 3.3.2. Surface Hydrophobicity (*H_0_*)

*H*_0_ can also indicate alterations in the protein structure [30]. Commonly, lower *H*_0_ of proteins indicated that those proteins’ structures were loosened and more highly exposed to a solution [30]. As shown in Figure 1d, a significant reduction (*p* < 0.05) in *H*_0_ was observed in DHBP after combining with PC, implying that PC prevented the exposure of hydrophobic plaques through hydrophobic interactions [31]. This phenomenon could be explained by the re-embedding of a hydrophobic group, preventing the binding to the fluorescent probe and causing *H*_0_ to decrease [29]. In addition, the *H*_0_ of DHBP–PC complexes (pH 7) was higher than that of DHBP–PC complexes (pH 9), implying that the unfolding state of the structure and the stability of DHBP–PC complexes at pH 7 was better than that of DHBP–PC complexes at pH 9 [30,32].

### 3.4. Fluorescence Spectroscopy Analysis

Fluorescence spectroscopy is often used to monitor the changes in a protein’s tertiary structure [33]. As shown in Figure 2, the emission fluorescence intensity of DHBP at around 347 nm appreciably decreased with the increased concentration of PC (Figure 2a,b). This result indicated that PC could combine with any or all tryptophan (Try) or tyrosine (Tyr) residues [34]. Additionally, the maximum emission fluorescence wavelength of DHBP–PC complexes was red shifted. In general, the red shift indicated that Try residues were more exposed to the solvent, or that they were being transferred to a more hydrophilic environment [35]. The red shift suggested that the *H*_0_ of DHBP–PC complexes decreased and illustrated that PC was connected to DHBP through hydrophobic interactions [31], which was consistent with the results of *H*_0_.

The fluorescence quenching data of DHBP at three different temperatures (298 K, 303 K, and 308 K) were analyzed using the Stern–Volmer equation. Generally, fluorescence quenching can be divided into static and dynamic quenching [36]. As shown in Figure 2c,d, the Stern–Volmer plots of DHBP–PC complexes formed at pH 7 and 9 all displayed excellent linearity (*R*^2^ > 0.99) at three temperatures, implying that the fluorescence quenching mechanism between DHBP and PC should, in theory, only be dynamic or static [36]. The calculated values of *K*_q_ [>10^13^ L/(mol·S)] were far more than the maximum diffusion collision quenching constant [2.0 × 10^10^ L/(mol·S)], which was one piece of evidence for the static quenching. Furthermore, the corresponding *K*_sv_ of DHBP–PC complexes (pH 7) decreased while the corresponding *K*_sv_ of DHBP–PC complexes (pH 9) increased with the rising temperature (Table 2). The results illustrated that the fluorescence quenching mechanism of the binding of PC to DHBP was static [37]. As shown in Figure 2e,f, the numbers of *n* were all close to 1, indicating the presence of one interaction site between PC and DHBP. The *K*_a_ values increased with the increasing temperature, indicating that the interaction between DHBP and PC was endothermic. The increase in temperature was beneficial to the combination of PC with DHBP.

The thermodynamic parameters were used to further elucidate the interaction of PC with DHBP. As shown in Table 2, the Δ*G* was negative, suggesting that the combination between DHBP and PC was spontaneous. The Δ*H* and Δ*S* were all positive, implying that hydrophobic forces were the main driving force in the interaction between DHBP and PC [36]. Furthermore, NaCl, sodium dodecyl sulfate (SDS), and urea were used in the DHBP–PC complexes (pH 7 and 9) to further explore the driving forces between DHBP and PC [12]. As shown in Appendix A, the addition of SDS significantly increased the fluorescence intensity of DHBP–PC complexes, implying the existence of hydrophobic interactions between DHBP and PC [12]. The results further proved that the interaction between DHBP and PC at pH 7 and 9 was mainly driven by hydrophobic forces.

### 3.5. Structural Analysis

#### 3.5.1. Free Sulfhydryl and Amino Groups

The free sulfhydryl and free amino group contents were used to monitor the attachment of PC to DHBP. As shown in Figure 3a, the reaction with PC at pH 7 and 9 resulted in a decrease in the free sulfhydryl group contents of DHBP. PC contains phenolic hydroxyl groups, which can be combined with the sulfhydryl groups of DHBP to change the structures of the protein [38]. Hence, the interaction between DHBP and PC could result in a decrease in the contents of free sulfhydryl groups. The decreased contents of free sulfhydryl groups of DHBP–PC complexes at pH 7 were in accordance with previous studies reporting that the sulfhydryl group content of pea protein isolates obviously decreased after the noncovalent binding of polyphenols [11]. Moreover, the reduction degree of the free sulfhydryl group content of DHBP–PC complexes formed at pH 9 was higher than that of DHBP–PC complexes formed at pH 7. An explanation for this phenomenon was that PC was oxidized to corresponding quinones under alkaline conditions, and the quinones easily reacted with free sulfhydryl or free amino groups on the DHBP to form covalent bonds [38]. Moreover, when the PC concentrations was 0–0.08 mmol/L, the free sulfhydryl group contents of DHBP–PC complexes at pH 7 were lower than those at pH 9.

As shown in Figure 3b, adding PC did not significantly change the free amine group contents of DHBP at pH 7. Meanwhile, compared with DHBP, the free amino group content of the DHBP–PC complexes at pH 9 decreased from 0.07 ± 0.01 mg/mL to 0.06 ± 0.00 mg/mL, suggesting that the free amino groups in DHBP would participate in the interaction with PC [33]. Furthermore, the result also indicated that the reactivity and binding strength of PC to free amino groups of DHBP increased. As reported, the oxidized polyphenols could react with nucleophilic groups of proteins, including the amino groups Try, histidine, cisterns, Tyr, methionine, and N-terminal proline [39]. Hence, the free amino group contents of the DHBP–PC complexes formed at pH 9 decreased. Furthermore, the reduction in the free amino group contents of DHBP–PC complexes at pH 9 might be attributed to the free amino groups in DHBP reacting with the carbonyl groups in PC to form the DHBP–PC glycation product [39]. Compared with the DHBP–PC complexes at pH 9, the free amino group contents of DHBP–PC complexes at pH 7 were lower, which might be dependent on the pH environment.

#### 3.5.2. FTIR

FTIR spectroscopy was applied to evaluate the structural changes in proteins. As shown in Figure 4, the FTIR of PC showed bands at 3379.72, 1161.47, 1522.29, 1443.25, and 1284.28 cm^−1^. As shown in Figure 4a, the DHBP at pH 7 had three typical peaks near 3300 cm^−1^ (amide A), 1638.84 cm^−1^ (amide I), and 1548 cm^−1^ (amide II) [38]. The 1538.84 cm^−1^ (amide II) of DHBP at pH 7 slightly shifted to a shorter wavelength after binding with PC. As shown in Figure 4b, the DHBP at pH 9 also had three typical peaks near 3283.42 cm^−1^ (amide A), 1633.87 cm^−1^ (amide I), and 1527.94 cm^−1^ (amide II). The complexes with added PC were changed in terms of amide I and amide II positions, suggesting the interaction between DHBP and PC [12,38]. The shifted amide I and amide II for DHBP–PC complexes indicated that the secondary structure of DHBP was altered after combination with PC.

#### 3.5.3. CD Analysis

Further insights into the impact of PC binding on DHBP structure were gained by examining the CD spectrum of DHBP. As shown in Figure 4c,d, the negative ellipticity of DHBP decreased after adding different concentrations of PC, especially in DHBP binding with PC at pH 9. The result suggested a change in the secondary structure of DHBP [40]. The structural changes in DHBP were reflected by four secondary structures (α-helixes, β-sheet, β-turn, and random coil). As shown in Appendix A, the native DHBP contained 18.99 ± 0.16% (pH 7) and 15.63 ± 0.09% (pH 9) α-helixes, 26.25 ± 0.07% (pH 7) and 23.89 ± 0.09% (pH 9) β-sheet, 22.90 ± 0.01% (pH 7) and 22.84 ± 0.06% (pH 9) β-turn, and 31.86 ± 23% and 33.08 ± 0.12% (pH 9) random coil. The α-helixes content of DHBP decreased, while the β-sheet content of DHBP increased after binding with PC at pH 7 and 9. Many studies suggested that the increase in the number of β-sheets and the reduction in the number of α-helixes resulted from the denaturation and unfolding of protein molecules [36,41]. Similarly, the combination of polyphenols and proteins promoted breakage of the hydrogen bonding network structures [40]. Hence, the changes in the secondary conformation of DHBP modified by PC indicated the unfolding of DHBP. The same result was reported in α-lactalbumin–hydroxy safflower yellow A, α-lactalbumin–neohesperidin dihydrochalcone, and α-lactalbumin–naringin dihydrochalcone complexes, which revealed that the binding of three chalconoids insignificantly reduced the α-helixes and β-sheet contents of α-lactalbumin, but induced a transition from α-helixes to β-sheet [9]. Meanwhile, insignificant differences in the β-turn and random coil contents were found between DHBP and DHBP–PC complexes. Furthermore, the sum content of α-helixes and β-sheet of DHBP–PC complexes at pH 7 was obviously higher than that at pH 9, while the sum content of β-turn and random coil of DHBP–PC complexes at pH 7 was lower than that at pH 9. The results suggested that DHBP–PC complexes at pH 7 were much more stable than DHBP–PC complexes at pH 7 [14].

### 3.6. Foaming Properties

As shown in Figure 5a,b, compared with DHBP, the FC and FS of DHBP–PC complexes formed at pH 7 and 9 significantly increased with increasing concentrations of PC (*p* < 0.05) and reached the maximum at 0.32 mmol/L PC. This finding indicated that the FC and FS of DHBP improved by binding with PC. The increased FC and FS were mainly due to the unfolding of the DHBP structure, which promoted the interaction between DHBP and air [42]. This result was consistent with the change in the fluorescence spectrum and *H_0_* of DHBP after combining with PC. In addition, the increase degree of FC and FS of DHBP–PC complexes at pH 7 was higher than that at pH 9. This finding implied that the unfolding degree of DHBP–PC complexes at pH 7 was better than that at pH = 9.

### 3.7. Emulsifying Property

As shown in Figure 5c, the EAI of DHBP–PC complexes was higher than that of the DHBP. In addition, the EAI of DHBP–PC complexes increased with the increased concentrations of PC (except for 0.64 mmol/L). A possible reason was that the combination of DHBP and PC changed the spatial structure and *H*_0,_ thus affecting the ability of the DHBP to adsorb to the oil–water surface [32]. Interestingly, the ESI of DHBP–PC complexes decreased with the increased concentrations of PC (in Figure 5d). There were some factors affecting the stability of the protein emulsions, such as interface tension, three-phase contact angle, particle size, and structure of protein and polyphenol. Additionally, the EAI and ESI of DHBP–PC complexes formed at pH 7 were higher than those at pH 9. The higher EAI of DHBP–PC complexes formed at pH 7 might be attributed to higher *H*_0_, which could enhance the binding ability with the oil droplets [38,42]. Meanwhile, the higher ESI of DHBP–PC complexes formed at pH 7 might be significantly associated with higher zeta potential [38].

### 3.8. Antioxidant Activity

As shown in Figure 6a, the DPPH radical scavenging ability of DHBP was weak. The DPPH scavenging ability of DHBP–PC complexes formed at pH 7 and 9 significantly increased with the increasing concentrations of PC. The DPPH scavenging ability of DHBP–PC complexes formed at pH 7 was higher than that of pure PC. Meanwhile, the DPPH scavenging ability of DHBP–PC complexes formed at pH 9 was lower than that of pure PC. As shown in Figure 6b, the ABTS scavenging ability of DHBP–PC complexes formed at pH 7 and 9 significantly increased with increasing concentrations of PC, indicating that the phenolic hydroxyl groups were introduced into DHBP. However, the ABTS scavenging ability of DHBP–PC complexes was lower than that of the corresponding concentrations of pure PC. A possible reason is that the active hydroxyl groups of PC were partially occupied during the formation of complexes between DHBP and PC [31]. As shown in Figure 6c, the FRAP of DHBP significantly increased after interacting with PC. Nevertheless, the FRAP of DHBP–PC complexes at different PC concentrations was significantly lower than that of pure PC at corresponding concentrations (*p* < 0.05). The formation of hydrogen bonds between DHBP and PC led to the occupation of hydroxyl groups in PC [30,31]. Therefore, the antioxidant activity of DHBP–PC complexes was lower than that of PC. Furthermore, compared with the DHBP–PC complexes formed at pH 7, the reducing or antioxidant abilities of DHBP–PC complexes formed at pH 9 were lower, which might be due to the oxidation of phenolics, leading to partial impairment [26].

## 4. Conclusions

In the present study, the PC interacted with DHBP to form complexes at different pH values. The fluorescence spectrum analysis results demonstrated that PC could quench the fluorescence of proteins by static quenching. PC affected the secondary structure and changed the turbidity, particle size, zeta potential, surface hydrophobicity, and the contents of free sulfhydryl and amino groups of DHBP. Furthermore, the interaction of PC and DHBP improved the reducing or antioxidant abilities of DHBP. Moreover, compared with DHBP–PC complexes formed at pH 9, the DHBP–PC complexes formed at pH 7 had a more stable structure, stronger antioxidant activity, and better foaming and emulsification properties. This study might provide a theoretical basis for improving the structural and functional properties of highland barley proteins. The study also had guiding value for the development and utilization of highland barley proteins and proanthocyanidins.

## Figures and Tables

**Figure 1 foods-12-00481-f001:**
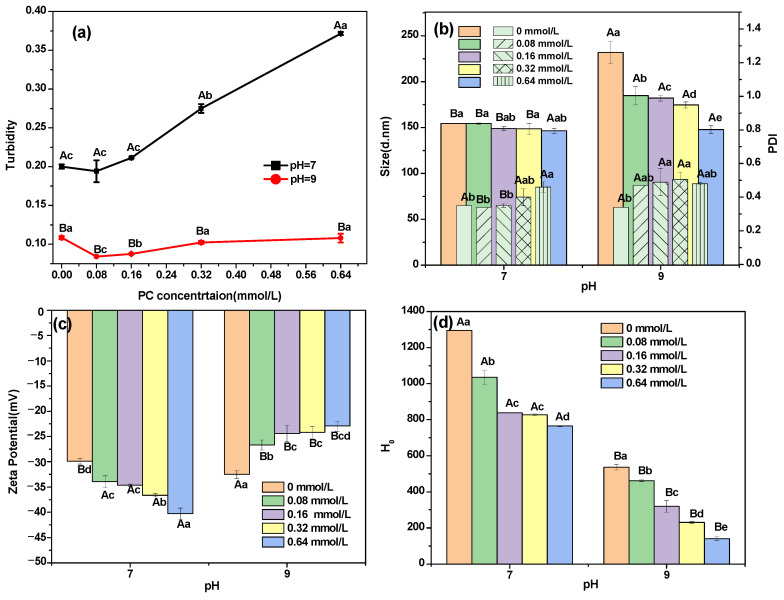
Turbidity (**a**), particle size (**b**), polydispersity index (**b**), zeta potential (**c**), and surface hydrophobicity (**d**) changes of decolorized highland barley protein (DHBP)–proanthocyanidin (PC) complexes formed at pH 7 and 9. ^a–e^ mean values under the same conditions with different letters, ^A,B^ mean values under different conditions with different letters, indicating significant differences (*p* < 0.05), respectively.

**Figure 2 foods-12-00481-f002:**
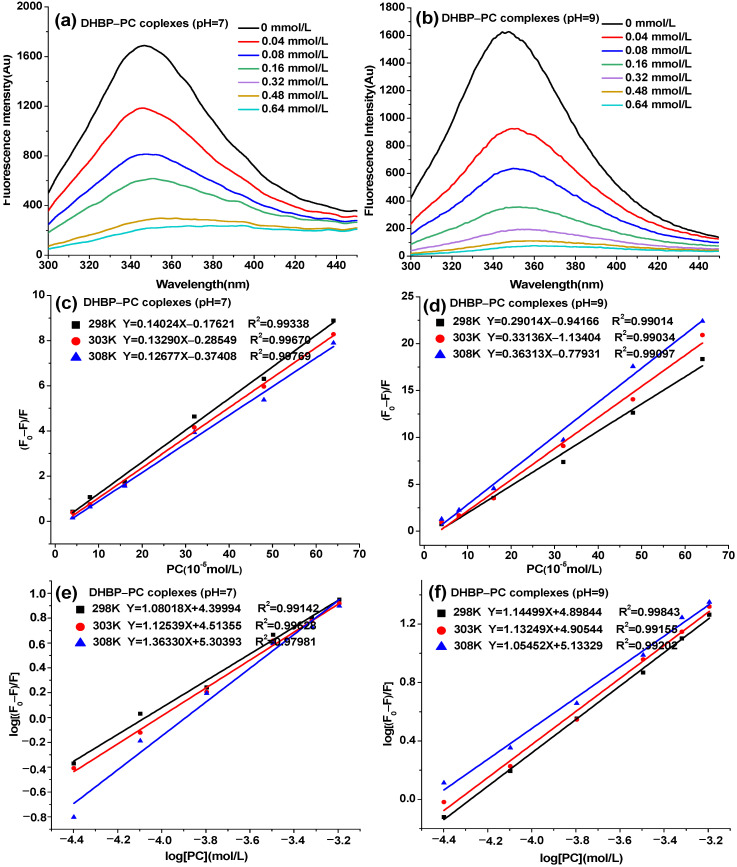
Fluorescence spectrum (**a**,**b**) and fluorescence quenching mechanism (**c**–**f**) analysis of decolorized highland barley protein (DHBP)–proanthocyanidin (PC) complexes.

**Figure 3 foods-12-00481-f003:**
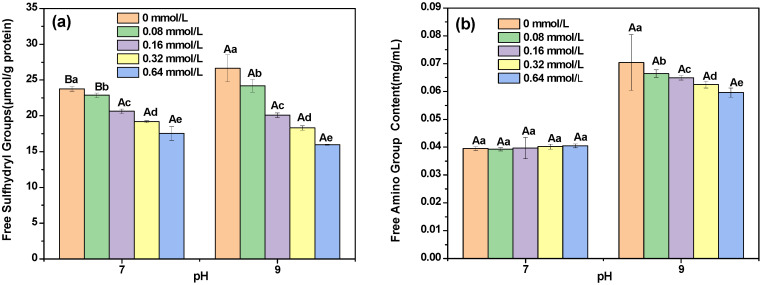
Analysis of free sulfhydryl content (**a**) and free amino group (**b**) content of decolorized highland barley protein (DHBP)–proanthocyanidin (PC) complexes. ^a–e^ mean values with different letters under the same conditions; ^A,B^ mean values with different letters under different conditions, indicating significant differences (*p* < 0.05), respectively.

**Figure 4 foods-12-00481-f004:**
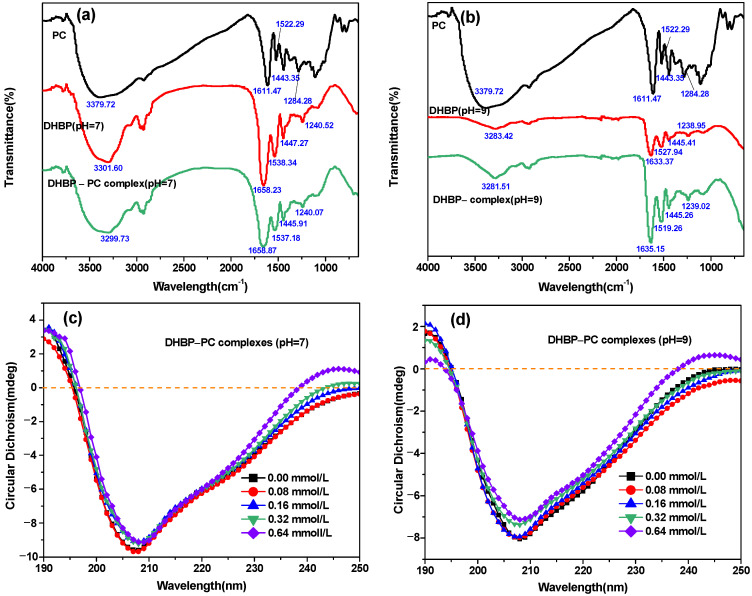
Fourier transform infrared spectra (**a**,**b**) and circular dichroism chromatography (**c**,**d**) analysis of decolorized highland barley protein (DHBP)–proanthocyanidin (PC) complexes.

**Figure 5 foods-12-00481-f005:**
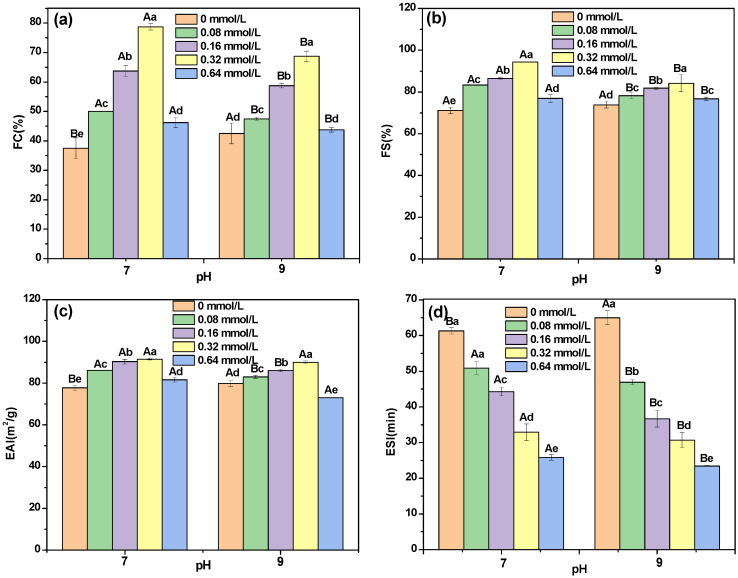
The foaming capacity (FC, (**a**)) and foaming stability (FS, (**b**)), emulsifying activity index (EAI, (**c**)), and emulsifying stability index (ESI, (**d**)) of decolorized highland barley protein (DHBP)−proanthocyanidin (PC) complexes. ^a–e^ mean values with different letters under the same conditions; ^A,B^ mean values with different letters under different conditions, indicating significant differences (*p* < 0.05), respectively.

**Figure 6 foods-12-00481-f006:**
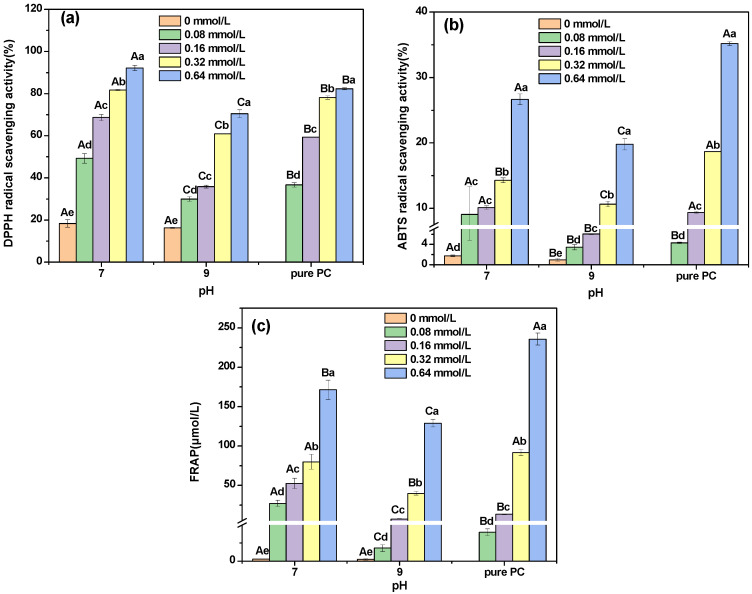
DPPH scavenging ability (**a**), ABTS radical scavenging ability (**b**), and FRAP reducing power (**c**) analysis of decolorized highland barley protein (DHBP)–proanthocyanidin (PC) complexes. ^a–e^ mean values with different letters under the same conditions; ^A–C^ mean values with different letters under different conditions, indicating a significant difference (*p* < 0.05), respectively.

**Table 1 foods-12-00481-t001:** Binding equivalent of decolorized highland barley protein (DHBP) and proanthocyanidin (PC) at different pH values (mg/g protein).

PC Concentration (mmol/L Protein)	DHBP–PC Complexes (pH = 7)	DHBP–PC Complexes (pH = 9)
0.08	12.05 ± 0.21 ^Ad^	10.00 ± 0.42 ^Bd^
0.16	25.87 ± 1.04 ^Ac^	22.05 ± 1.25 ^Bc^
0.32	59.18 ± 0.83 ^Ab^	54.36 ± 3.12 ^Bb^
0.64	135.04 ± 1.66 ^Aa^	117.57 ± 3.12 ^Ba^

^a–d^ mean values in the same column with different letters; ^A,B^ mean values in different columns with different letters, representing significant differences (*p* < 0.05), respectively.

**Table 2 foods-12-00481-t002:** Fluorescence quenching parameters of decolorized highland barley protein (DHBP)–proanthocyanidin (PC) complexes.

	DHBP–PC Complexes (pH = 7)			DHBP–PC Complexes (pH = 9)		
	298 K	303 K	308 K	298 K	303 K	308 K
*K*_sv_ (L/mol)	1.4024 × 10^4^	1.3290 × 10^4^	1.2677 × 10^4^	2.9014 × 10^4^	3.3136 × 10^4^	3.6313 × 10^4^
*K*_q_ (L/mol/s)	1.4024 × 10^12^	1.3290 × 10^12^	1.2677 × 10^12^	2.9014 × 10^12^	3.3136 × 10^12^	3.6313 × 10^12^
*K*_a_ (L/mol)	2.51 × 10^4^	3.26 × 10^4^	2.01 × 10^5^	7.91 × 10^4^	8.04 × 10^4^	1.36 × 10^6^
*n*	1.08018	1.12539	1.13633	1.14499	1.13249	1.05452
Δ*G* (kJ/mol)	–27.52			–28.89		
Δ*H* (kJ/mol)	159.51			41.48		
Δ*S* (kJ/mol/K)	0.62			0.23		

## Data Availability

All data generated or analyzed during this study are included in this published manuscript.

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
