# Peer review of "Impacts of Proanthocyanidin Binding on Conformational and Functional Properties of Decolorized Highland Barley Protein"

_foods, 2023, doi:10.3390/foods12030481_

Round 1
Reviewer 1 Report
As mentioned by the Author in the MS:
"It is rich in proteins, fibers, vitamins, β-glucans, arabinoxylans, poly-33 phenols and flavonoids [2-4]. Furthermore, it contains more lysine and threonine, which 34 are considered limiting amino acids [5]."
Q: Cereal proteins are deficient in lysine and rich in glutamic acid. If highland barley is high in lysine, the value must be mentioned in the statement.
Q: Modern food industry needs pulse proteins for the production of gluten-free as well as texturized vegetable meat analogs, the author should justify the application of barley proteins in the food industry or the alcoholic beverage industry.
The author can see below their own statements:
"For example, soy proteins accompanying anthocyanins could change the secondary 55 structure, improve the foaming properties and emulsifying properties of soy proteins [10]. 56 The surface hydrophobicity significantly reduced, and the emulsification, foaming, di-57"
The Author should clarify the use of H2O2 at the time of protein isolates preparation.
"Then, hydrogen peroxide (3%, v/v) was added to the suspension and 88 kept at 40°C for 60 min."
Author evaluated the hypothesis with very high-end machines, like CD, FTIR, Flourescent Spectrometry, however, very basic analysis is missing, the functionality of these proteins,
Foaming capacity
Foaming stability
Emulsification capacity
Emulsification stabilility
These parameters must be evaluated and corelated with FTIR, CD, Flourescent data etc.,
English language improvement is required. Grammer also need critical reading.
Author Response
Dear reviewer,
Thank you very much for the comments. We have checked the manuscript carefully and revised it according to the comments. The detailed changes of manuscript are listed below:
Response to reviewer 1:
As mentioned by the Author in the MS:
"It is rich in proteins, fibers, vitamins, β-glucans, arabinoxylans, polyphenols and flavonoids [2-4]. Furthermore, it contains more lysine and threonine, which are considered limiting amino acids [5]."
Comment 1: Cereal proteins are deficient in lysine and rich in glutamic acid. If highland barley is high in lysine, the value must be mentioned in the statement.
Response:Highland barley contains more lysine (0.360 g/100g) and threonine (0.360 g/100g ), which are considered limiting amino acids. Please see in line 35 in revised manuscript.
Comment 2: Modern food industry needs pulse proteins for the production of gluten-free as well as texturized vegetable meat analogs, the author should justify the application of barley proteins in the food industry or the alcoholic beverage industry.
The author can see below their own statements:
"For example, soy proteins accompanying anthocyanins could change the secondary structure, improve the foaming properties and emulsifying properties of soy proteins [10]. 56 The surface hydrophobicity significantly reduced, and the emulsification, foaming, di-57"
Response:The data about the structural and functional properties of the highland barley protein are lacking; most researchers focused on their derivatives, that is, the peptides. Besides, natural highland barley proteins cannot meet different needs of food processing. For example, a large amount of wheat flour and gluten powder were often added in the processing of highland barley noodles due to highland barley protein cannot form gluten network structure.
In our previous studies, we studied the effect of salidroside on the structure and function of highland barley protein (including total highland barley protein, gliadin and glutenin). The results showed that the foaming, emulsifying and antioxidant activities of highland barley protein significantly increased after the interaction of salidroside and highland barley protein (line 66-69).
Comment 3: The Author should clarify the use of H2O2 at the time of protein isolates preparation. "Then, hydrogen peroxide (3%, v/v) was added to the suspension and kept at 40°C for 60 min."
Response:The color of highland barley protein extracted using alkali and acid precipitation was brown, which might be due to the combination of a variety of phenolic substances under alkaline conditions. Therefore, the barley protein was decolorized to reduce interference. In our previous research, we used single factor test and orthogonal test to screen and optimize the decolorization conditions of highland barley protein. Finally, the optimal decolorization conditions were obtained: 3% hydrogen peroxide (v/v) was the highland barley protein suspension (pH 11) and kept at 40°C for 60 min. The specific optimization process and parameters can be seen the published reference: Effects of decolorization on aggregation behavior of highland barley proteins: Comparison with wheat proteins. Food Research International, 2022. Please see in line 90 in revised manuscript.
Comment 4: Author evaluated the hypothesis with very high-end machines, like CD, FTIR, Flourescent Spectrometry, however, very basic analysis is missing, the functionality of these proteins: foaming capacity, foaming stability, emulsification capacity, emulsification stability
These parameters must be evaluated and correlated with FTIR, CD, Flourescent data etc.
Response:The foaming capacity, foaming stability, emulsification capacity, emulsification stability of DHBP and DHBP-PC complexes have been evaluated and correlated with FTIR, CD, Flourescent data in revised manuscript. The determination method of foaming capacity, foaming stability, emulsification capacity, emulsification stability see in line 170-176 (method) and line 395-419 in revised manuscript.
Comment 5: English language improvement is required. Grammer also need critical reading.
Response:We have asked help from the Editorbar Language Editing (Beijing, China). The English writing of our manuscript was carefully edited by a native English speaker thoroughly. Please see the CERTIFICATE OF LANGUAGE EDITING (in the attachment).

Reviewer 2 Report
I have reviewed the submitted manuscript (foods-2138364; Impacts of bounding proanthocyanidin on conformational and functional properties of decolorized highland barley protein). Reviewed manuscript is dedicated to the experimental study and investigate the impacts of interaction between proanthocyanidin (PC) and decolorized highland barley protein (DHBP) at pH 7 and 9 on the functional and conformational changes in DHBP. Paper is well structured and all experimental procedures are clearly described. Manuscript contains some interesting observations, however, there are several additional comments regarding this manuscript:
Line 53: conbination of proteins → remove extra space
Line 92: isoelectric point→ ???
Line 181: Statistical analysis→ You didn’t compare results of pH=7 and pH=9. Why?
Line 236: In brief, the DHBP–PC complexes at pH 7 were more stable than DHBP–PC complexes at pH 9→ You didn’t compare and analysis results of pH=7 and pH=9 statistically. How can proof that?
Author Response
Dear reviewer,
Thank you very much for the comments. We have checked the manuscript carefully and revised it according to the comments. The detailed changes of manuscript are listed below:
Response to reviewer 2:
I have reviewed the submitted manuscript (foods-2138364; Impacts of bounding proanthocyanidin on conformational and functional properties of decolorized highland barley protein). Reviewed manuscript is dedicated to the experimental study and investigate the impacts of interaction between proanthocyanidin (PC) and decolorized highland barley protein (DHBP) at pH 7 and 9 on the functional and conformational changes in DHBP. Paper is well structured and all experimental procedures are clearly described. Manuscript contains some interesting observations, however, there are several additional comments regarding this manuscript:
Comment 1: Line 53: conbination of proteins → remove extra space
Response:Thank you for your suggestion. We have removed the extra space in line 53 in revised manuscript.
Comment 2: Line 92: isoelectric point→ ???
Response:Thank you for your suggestion. The isoelectric point of HBP was 4.5. We have deleted the redundant "isoelectric point" in line 93.
Comment 3: Line 181: Statistical analysis→ You didn’t compare results of pH=7 and pH=9. Why?
Response:We have added comparative analysis of turbidity (line 217-219), particle size (232-234), zeta potential (line 253-254), surface hydrophobicity (line 263-266), free sulfhydryl group (332-333) and amino groups content (345-347), secondary structure content (line 384-388), foaming (402-404) and emulsification properties (414-419), and antioxidant activity (421-440) of DHBP–PC complexes at pH 7 and pH 9. Please see in revised manuscript.
Comment 4: Line 236: In brief, the DHBP–PC complexes at pH 7 were more stable than DHBP–PC complexes at pH 9→ You didn’t compare and analysis results of pH=7 and pH=9 statistically. How can proof that?
Response: The absolute zeta potential of DHBP–PC complexes formed at pH 7 increased significantly with the increasing concentration of PC. The absolute zeta potential of DHBP bound PC at pH 7 was higher than that at pH 9. A higher zeta potential value means more stable of protein solution. Therefore, the DHBP–PC complexes at pH 7 were more stable than DHBP–PC complexes at pH 9. Besides, the PDI, surface hydrophobicity, stable secondary structure content (α-helixes and β-sheet) of DHBP–PC complexes at pH 7 were all higher than those at pH 7, indicating that the stability of DHBP–PC complexes at pH 7 was better than DHBP–PC complexes at pH 9. Please see in line 232-234, 263-266, 387-319 in revised manuscript for relevant content.
Round 2
Reviewer 2 Report
No Comment